# Influence of V and Zn in FeCrCuMnTi High-Entropy Alloys on Microstructures and Uniaxial Compaction Behavior Prepared by Mechanical Alloying

**Subbarayan Sivasankaran** [1,*], **Fahad A. Al-Mufadi** [1] and **Hany R. Ammar** [1,2]

1 Department of Mechanical Engineering, College of Engineering, Qassim University, Buraydah 51452, Saudi Arabia; almufadi@qec.edu.sa (F.A.A.-M.); hanyammar@qec.edu.sa (H.R.A.)
2 Metallurgical and Materials Engineering Department, Faculty of Petroleum and Mining Engineering, Suez University, Suez 43721, Egypt
* Correspondence: sivasankaran@qec.edu.sa

**Abstract:** The densification behavior of FeCrCuMnTi (HEA1), FeCrCuMnTiV (HEA2), and FeCrCuMnTiVZn (HEA3) equiatomic high-entropy alloys (HEAs) was explored using different uniaxial quasi-static controlled compaction (1 mm/min). These HEAs were synthesized by mechanical alloying (MA, speed: 300 rpm, BPR: 10:1, time: 25 h). Various phase formations, structural characteristics (crystallite size, lattice strain, and lattice constant), thermo-dynamic calculations, powder surface morphologies, detailed microstructural evolutions, and chemical compositions were examined using X-ray diffraction, high-resolution scanning electron microscopy, and high-resolution transmission electron microscopy. The XRD results revealed the formation of multiple solid solutions (FCC, BCC, and HCP) due to the variation in entropy, and the presence of high-strength elements (Cr, Ti, and V) in the developed HEA alloys. The synthesized powders were consolidated into bulk green samples with different compaction pressures starting from 25 to 1100 MPa under as-milled and milled under stress recovery conditions (150 °C, 1 h). The incorporation of V in the FeCrCuMnTi HEA resulted in improved densification due to a greater reduction in particle size, and high configurational entropy. Furthermore, the stress-recovered powder samples produced more relative density owing to the elimination of lattice strain. Several linear and non-linear compaction models were applied to predict densification behavior. The non-linear Cooper and Eaton model produced the highest regression coefficients compared to the other models.

**Keywords:** high-entropy alloys; microstructures; powder particle size; densification; compaction models

## 1. Introduction

High-entropy alloys (HEAs) have recently attracted all materials scientists and materials engineers as several new alloys could be developed through this concept. HEAs exhibit extensive properties owing to the mixing of high-configurational entropy and solid solutions with severe lattice distortion which cannot be obtained by conventional alloys [1]. In conventional alloys, one metallic element acts as a major constituent (solvent atom), and the other metallic elements (solute atoms) contribute to the improvement of their properties [2–4]. However, it is very difficult to achieve the expected properties (strength, wear and corrosion resistance, high temperature withstanding properties, magnetic properties, superconductivity, and anti-radiation properties) in conventional alloys. However, these properties can be achieved in HEAs recommended for use in aircraft, space, defense, automotive, molding tool and die making parts, electrical and electronic parts, chemical engineering industries, and so on [5]. In HEAs, a minimum of five metallic elements can be added in equal molar ratios or non-equal molar ratios exhibiting improved properties [6,7]. Major futuristic characteristics such as cocktail effect, severe lattice distortion effect, high-entropy effect, and sluggish effect from the HEAs could be obtained, which are the main contributors to enhancing the properties [8].

Cantor et al. [9] explored the microstructural characteristics of FeCrMnNiCo multicomponent alloys. The authors have manufactured alloys through casting and melt spinning techniques. These alloys exhibited a single FCC high stability phase structure and produced extensive strength with more ductility at room temperature. It was found that the observed strength was greater than that of steel. Fazakas et al. [10] investigated the influence of the addition of Fe in AlTiNiCuFe HEAs prepared by arc melting under an argon atmosphere. The manufactured alloys were heat-treated by varying the temperature to examine the effect of aging on their mechanical behavior. The authors found higher ductility with high strength in these alloys. In addition, the authors explored that $Al_{22.5}Ti_{22.5}Ni_{20}Cu_{20}Fe_{15}$ HEA heat-treated at 973 K exhibited outstanding deformation behavior during compression. Hemphill et al. [11] examined the fatigue behavior of the AlCoCrCuFeNi HEA manufactured by the arc melting method. The authors found that the developed AlCoCrCuFeNi HEAs exhibited high fatigue resistance even at high stress, which was considerably higher than that of steel, bulk metallic glass, and titanium alloy.

Schuh et al. [12] studied the mechanical and thermal stabilities of CoCrFeMnNi HEAs manufactured by arc melting. The CoCrFeMnNi HEAs were plastically deformed by high-pressure torsion (HPT) to refine their internal structure. The authors reported that the manufactured HEAs produced an ultimate strength of approximately 2 GPa with a hardness of approximately 5 GPa. AlCoCrCuNiFeZn high-entropy alloy was synthesized by Babu et al. [13] by a mechanical alloying (MA) route; which is a solid-state powder metallurgy process. It was found that this alloy exhibited a single-phase BCC structure with excellent chemical resistance, and the authors achieved a stable alloy with a crystallite size of 15 nm. It was found that the produced alloy retained its properties even at 1023 K and produced a hardness strength of approximately 8 GPa with a modulus of elasticity of 0.16 TPa. Maulik and Kumar [3] developed AlFeCuCrMg HEAs processed via MA. The synthesized AlFeCuCrMg HEA powders exhibited a larger BCC structure with a small amount of FCC structure. It was found that the incorporated Mg elemental powders enhanced the formation of a stable single BCC structure even at 500 °C.

Fang et al. [14] studied the microstructure and mechanical behavior of AlCrFeNiCoC HEAs synthesized by the MA process and consolidated via spark plasma sintering (SPS). The authors achieved a supersaturated solid solution after 38 h of MA with a regular crystal system. Furthermore, the alloy exhibited a compressive strength of 2.1 GPa with a hardness strength of approximately 6 GPa strength, which is 14 times higher than that of conventional Al-based high strength alloys. Chen et al. [15] examined the mechanical behavior of AlCoNiCrFe HEAs processed by MA and reported that the produced alloy retained its crystallite size at the nano-level even after the sintering process. Wang et al. [16] developed NiCoCuFeCr HEAs in which the Ni-to-Cr content ratio was determined. These HEAs were synthesized using MA followed by SPS. The authors found that the formation of a single-phase FCC structure increased with increasing Ni/Cr ratio. In addition, the samples produced a tensile yield strength of 1 GPa with a considerable amount of ductility. Avila-Rubio et al. [17] developed three equiatomic AlCoFeNiZn, AlCoFeNiMoTi, and AlCoFeNiMoTiZn HEAs and the results indicated that the incorporation of Zn atoms promoted the BCC structure, decreased the crystallite size much, and introduced more lattice distortions. Pan et al. [18] synthesized FeCoNi(MoW)x (x = 0.2, 0.3, 0.5) HEAs using the vacuum arc melting method. The bulk samples exhibited improved ultimate compressive strength with increasing Mo, and W contents in the alloys. However, the formation of μ phase (due to Mo, and W) decreased the ductility and increased the brittleness. Son et al. [19] developed a CoCrFeMnNi HEA through a vacuum induction melting furnace and the results showed that the CoCrFeMnNi HEA exhibited improved antifouling properties compared to 304 stainless steel. The mechanical behavior of most HEAs is affected by the presence of hydrogen embrittlement in the grain boundaries, which was recently studied by Zhu et al. [20] who developed FeCrNiMnCo HEA. Daryoush et al. [21] studied the crystallization kinetics and amorphization of both equiatomic and non-equiatomic AlFeCuZnTi HEAs synthesized by high-energy ball milling. Liang et al. [22] investigated

the phase stability performance of aged (Co1.5FeNi)90Ti6Al4 HEA produced through vacuum arc melting followed by aging at 1073 K. Xia et al. [23] synthesized FeCrCuTiV HEAs using two different methods (through vacuum arc melting and laser melting deposition), and the results demonstrated that the HEA produced from laser melting deposition retained the grains under fine size. Furthermore, the HEA produced BCC and FCC phases. Liu et al. [24] developed a non-equiatomic FeMnCrNiAl HEA via arc melting followed by cold rolling and annealing. The sample exhibited a heterogeneous structure (matrix and precipitate phases) which promoted strength and ductility. Garlapati et al. [25] studied the thermal stability of Al content in AlxCoCrFeNi HEA powders at low and intermediate temperatures. A mixture of FCC and BCC structures was also observed. Ammar et al. [26] investigated the influence of milling time on $Al_{0.3}CrFeNiCo_{0.3}Si_{0.4}$ HEAs (1, 5, 15, and 25 h). The results revealed that the formation of a solid solution with BCC and FCC structures was obtained in a 25 h HEA sample. Alshataif et al. [1,27] reported the entropy effect on non-equimolar ratios of $Cr_{0.26}Fe_{0.24}Al_{0.5}$, $Cr_{0.21}Fe_{0.20}Al_{0.41}Cu_{0.18}$, $Cr_{0.15}Fe_{0.14}Al_{0.30}Cu_{0.13}Si_{0.28}$, and $Cr_{0.14}Fe_{0.13}Al_{0.26}Cu_{0.11}Si_{0.25}Zn_{0.11}$ HEAs prepared by MA. The literature explains that HEAs can be easily synthesized by MA without any detrimental effects as obtained from the arc melting route. Mane et al. [28] synthesized CoFeNi, CoFeNiCr, CoFeNiCrMn, and CoFeNiCrMnAl alloys and studied the densification rate with an increase in the number of elements during non-isothermal sintering. The results explained that the densification rate decreased with an increase in configurational entropy. Uniaxial compaction is an important process for producing bulk green samples before sintering in powder metallurgy industries [29]. The chemical composition, powder particle size, particle size distribution, powder surface morphology, applied compaction pressure, lubricant, and die wall friction, etc., are the major influencing parameters affecting the densification of bulk green products before sintering [30]. The densification behavior and its mechanisms have been studied by several authors in the past decades using various empirical models [26,31,32]. However, the influence of high configurational entropy, various phase formations, the presence of multi-phase solid solutions, and structural refinement on the densification behavior of HEAs have not been studied. Hence, the present study was conducted. The metallic element of Zn has a crystalline nature possessing high ductility and malleable properties operating at elevated temperatures. Furthermore, most parts of the die-casting industry are produced through Zn based alloys. The metallic element of V also possesses high strength with good ductile characteristics acting as an effective alloying element in more steel products. Therefore, in the present work, the incorporation of Zn and V was introduced in an equiatomic FeCrCuMnTi high entropy alloy. The main objectives of the present study are to: synthesize equiatomic FeCrCuMnTi, FeCrCuMnTiV, and FeCrCuMnTiVZn HEAs, study the structural characteristics (phase formations, crystallite size, lattice strain, and lattice constant), powder morphology examination, powder particle size analysis, study the various powder densities, and investigate the densification behavior of HEA powders under different compaction pressures (25–1100 MPa) under as-milled and milled with stress-recovered conditions.

## 2. Materials and Methods

Three equiatomic HEA powders of FeCrCuMnTi, FeCrCuMnTiV, and FeCrCuMnTiVZn were synthesized by MA. Metallic elemental powders (Fe, Cr, Cu, Mn, Ti, V, and Zn) with >99.5% purity as per chemical composition (Table 1) were purchased from M/S Nanografi, Germany. The average particle size of the as-received elemental powders was less than 44 μm (−325 mesh size). The powders were weighed in an electronic balance at an equiatomic ratio and charged into a high-energy ball mill for the MA process. The MA was carried out at a speed of 300 rpm, BPR of 10:1, and milling time of 25 h [26] in a Pulverisette 5/2 classic line (two stations) ball mill under toluene (wet milling to minimize cold welding). Tungsten carbide (WC) vials (250 mL capacity of each vial) and WC balls (Ø10 mm) were used. A total of 100 milling cycles were set and the MA was carried out automatically with 15 min forward, 15 min pause, 15 min reverse, and 15 min pauses to

minimize the heating effect. Some of the HEA powders were stress-relieved at 150 °C for 1 h in a vacuum tube furnace (M/S N Nabatherm, Germany). LECO CS 744 and ONH 836 gas analyzers were used to determine the presence of O, C, H, N, and S contents in the synthesized HEAs after 25 h MA in toluene (Table 1). The synthesized equiatomic HEA powders were scanned using an X-ray diffractometer (XRD, Empyrean, Malvern Panalytical, source: CuKa = 1.54 Å) at a scanning speed of 0.6°/min with a step of 0.01°, and X'Pert High Score Plus software was used to examine the obtained XRD peaks. The Debye–Scherrer formula (Equation (1)) was used to determine crystallite size [33].

$$ t = \frac{k\lambda}{\beta cos\theta} \tag{1} $$

where '*t*' is the crystal size, '$\lambda$' is X-ray wave-length, '$\theta$' is Bragg's angle, and '$\beta$' is full-width half-maximum. Instrumental broadening was corrected using Equation (2).

$$ \beta_{hkl} = \left[ (\beta_{hkl})^2_{Measured} - (\beta_{hkl})^2_{Instrumental} \right]^{1/2} \tag{2} $$

Apreo FEG-HR-SEM (30 keV, 1.3 nm resolution at 1 keV) was used to examine the powder surface morphology of the as-received powders and milled powders. The lattice constant for the cubical phases ('*a*' for BCC and FCC) and HCP phase ('*a*' and '*c*') was calculated based on Bragg's law. The strongest peaks were used to determine the lattice constants as per Equations (3)–(5) [33,34]:

For cubical systems,

$$ a = \frac{\lambda}{2sin\theta} \sqrt{h^2 + k^2 + l^2} \tag{3} $$

For the HCP system,

$$ \frac{1}{d^2_{hkl}} = \left( \frac{4}{3} \left( h^2 + k^2 + hk \right) + l^2 \left( \frac{a}{c} \right)^2 \right) \frac{1}{a^2} \tag{4} $$

For ideal, $c = 1.633a$; $a^2 = d^2_{hkl} \left( \frac{4}{3} \left( h^2 + k^2 + hk \right) + 0.375l^2 \right)$ (5)

The chemical composition of the milled powders was checked using an EDAX detector at three different locations and the average was used for interpretation. HRTEM of JEOL 3010 was used to investigate the synthesized HEA powders. The powders were dissolved in an ethanol solution, poured into a copper grid, and placed inside the machine. A Zetasizer (Malvern Panalytical) was used to measure the powder particle size and distribution. The apparent density and tap density of the synthesized powders were measured using a tap density meter (M/S DongGuan HongTuo Instrument Co., Ltd., Dongguan, China) according to ISO3923 and ETD1020 standards, respectively. The true density of the powder was also determined using a pycnometer (true density meter, DongGuan HongTuo Instrument Co., Ltd., Dongguan, China) as per the DIN51057 standard. The densification behavior of powders (as-milled and with stress-relieved conditions) was experimentally investigated at different compaction pressures from 25 to 1100 MPa. Uniaxial compaction was performed in cylindrical die-sets made of H13 steel at 1 mm/min in a universal testing machine (MTS, New York, NY, USA). The diameter of the inner die was 15 mm. Approximately 10 g of milled powder was poured into the die-set for compaction. A solid lubricant (zinc stearate) mixed with lubricating oil was used inside the die wall and punch before pouring the powders. Four replicates were performed at each compaction pressure and the average was used to examine the densification performance. The density of the green compacts/pellets was determined using Archimedes' principle. Several empirical formulae were used to predict the densification behavior and determine the best model for predicting the relative density of the developed powders. Figure 1 shows a schematic representation of the synthesis of nanostructured HEA powders by MA and the densification behavior used in the present study.

**Table 1.** Chemical composition, the crystallite size, lattice strain, and lattice constant of three developed high entropy alloys.

| Alloy Code | Alloy Composition (Atomic Fraction of Each Element) | LECO CS 744 and ONH 836 Analyzers (wt.%) | | | | | Crystallite Size (t), nm | | | Lattice Strain, % | | | Lattice Constant, nm | | | |
|---|---|---|---|---|---|---|---|---|---|---|---|---|---|---|---|---|
| | | O | C | H | N | S | BCC | FCC | HCP | BCC | FCC | HCP | BCC 'a' | FCC 'a' | HCP 'a' | HCP 'c' |
| HEA1 | FeCrCuMnTi (0.2) | 0.68 | 0.45 | 0.08 | 0.09 | 0.005 | 18 | 9 | 31 | 0.97 | 0.95 | 1.5 | 0.485 | 0.361 | 0.296 | 0.483 |
| HEA2 | FeCrCuMnTiV (0.167) | 0.75 | 0.62 | 0.09 | 0.10 | 0.004 | 24 | 10 | 48 | 0.96 | 0.92 | 2.9 | 0.491 | 0.363 | 0.300 | 0.490 |
| HEA3 | FeCrCuMnTiVZn (0.143) | 0.82 | 0.74 | 0.12 | 0.09 | 0.001 | 14 | 6 | 22 | 1.1 | 0.98 | 5.6 | 0.502 | 0.365 | 0.308 | 0.503 |

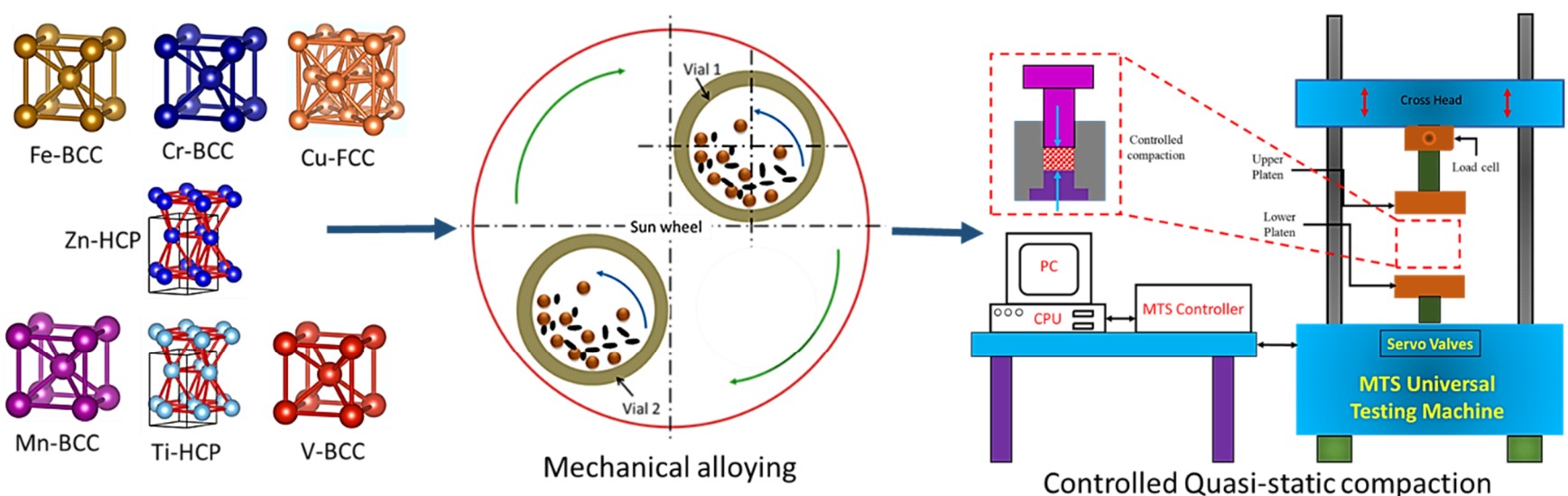

**Figure 1.** Schematic showing the present research work to synthesize and investigate the densification behavior of three high entropy alloys.

## 3. Results

### 3.1. Phase Evolutions and Structural Characterization Using XRD

Figure 2a shows the XRD patterns of FeCrCuMnTi, FeCrCuMnTiV, and FeCrCuMn-TiVZn HEAs after 25 h of MA. The results revealed that three phases, namely, BCC, FCC, and HCP were obtained in all the developed HEAs. More peak broadening (full-width half-maximum, FWHM) and a lower peak intensity were obtained for all the samples indicating the formation of a multi-phase solid solution. This was attributed to more crystallite fragmentation, severe lattice distortion, presence of lattice strain, cocktail effect, high entropy effect, and sluggish effect occurring in the MA process [8,27,35]. Figure 2b shows the XRD patterns of 0 h blended samples for comparing the solid solution formation in the developed nanocrystalline HEAs. As shown in Figure 2b that more intense peaks corresponding to incorporated elements were observed indicating no solid solution formation. However, after 25 h MA, more elemental peaks disappeared; the peak intensity decreased drastically, and; the peak width increased considerably confirming the formation of a solid solution. The value of FWHM in each phase and peak intensity values varied with increasing entropy (number of elements) and depended on the melting point of the incorporated elements. Among the three developed HEAs, FeCrCuMnTiVZn HEA3 exhibited a higher FWHM due to the easy dissolution of low-melting Zn atoms and hence this HEA3 produced more crystallite size reduction and more lattice strain (Table 1) compared to other HEAs. The order of dissolution of atoms depends on the melting point of each incorporated element [1]. The expected dissolution series of the three developed HEAs were Cu→Mn→Fe→Ti→Cr, Cu→Mn→Fe→Ti→Cr→V, and Zn→Cu→Mn→Fe→Ti→Cr→V for FeCrCuMnTi, FeCrCuMnTiV, and FeCrCuMnTiVZn HEAs, respectively. The incorporated elements of Fe, Cr, and Cu almost merged in all samples and produced a major FCC phase.

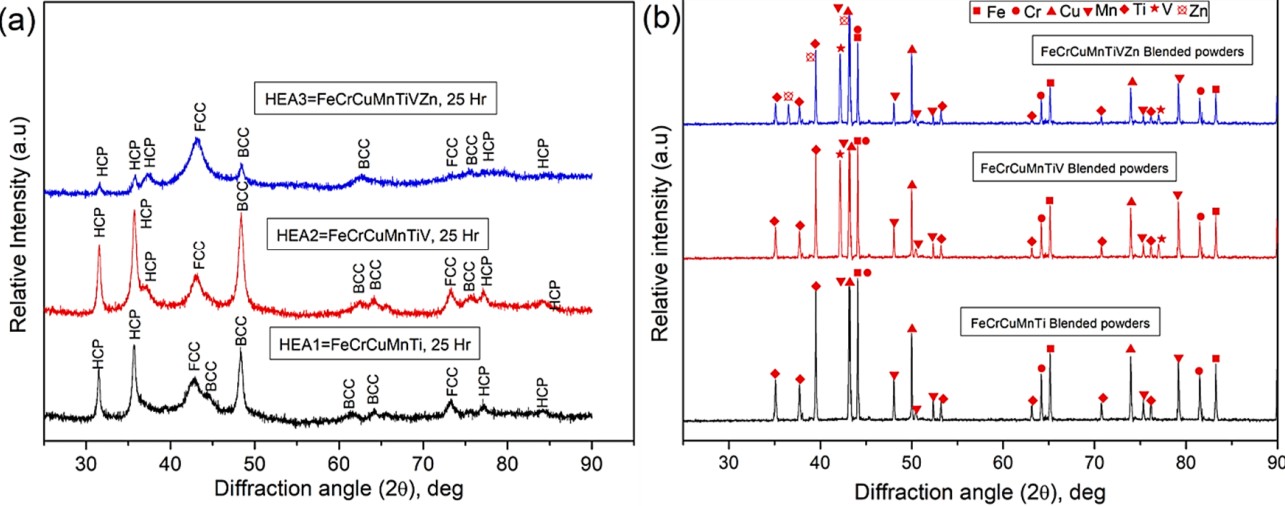

**Figure 2.** XRD patterns of different equiatomic high-entropy alloys powders of FeCrCuMnTi (HEA1), FeCrCuMnTiV (HEA2), and FeCrCuMnTiVZn (HEA3) synthesized by MA: (**a**) 25 h; (**b**) 0 h blended samples for comparison.

The incorporation of Mn atoms promoted the formation of the BCC phase in HEA1, and a higher peak intensity of the BCC phase was observed in HEA2 owing to the incorporation of V atoms. Moreover, the incorporation of Zn atoms suppressed the formation of the BCC phase and promoted the formation of the FCC phase in HEA3. The HCP phase formation in HEA1 and HEA2 was attributed to the incorporation of Ti atoms and this HCP phase was suppressed in HEA3 because Zn atoms formed more FCC phases. The calculated lattice constants (Equations (3)–(5), Table 1) in each phase of all the samples were determined and it was observed that the lattice constant increased due to high config-

urational entropy in addition to structural refinement. For instance, the lattice constants for the FCC phase were 0.361, 0.363, and 0.365 nm for HEA1, HEA2, and HEA3, respectively indicating that more structural refinement and lattice distortion occurred in HEA3.

### 3.2. Microstructural Examinations Using HRSEM and HRTEM

The microstructural examinations for the three developed equiatomic HEAs were carried out using HRSEM and HRTEM. Figure 3 shows the HRSEM microstructure of as-received elemental powders showing an irregular shape of Fe, platelet shape of Cr, different sizes in the spherical shape of Cu, different sizes in the polygonal shape of Mn, irregular cum polygonal shape of Ti, irregular flaky shape of V, and spherical shape with satellite shape of Zn. All as-received powder particles had different shape morphologies, sizes before processing and the average particle size was less than 44 μm. The top right inset of each image in Figure 3 explains the powder surface morphology of the as-received powders at higher magnification. Further, HRSEM-EDS spot analyses of the as-received individual powder particles were also carried out to ensure the purity and confirm the name of the metallic elements, which are illustrated on the right side of Figure 3. Figure 4 shows the HRSEM powder surface morphologies of 25 h MAed HEAs powders exhibiting an almost equiaxed particle shape representing the attainment of a steady-state [27].

Some agglomerated large powder particles were observed in all the samples owing to the MA process and the high-entropy effect. Severe plastic deformation, cold-welding, and fracturing of powder particles occur in the MA process leading to the production of agglomerated particles during the domination of cold-welding and fine particles during the domination of fracturing. These phenomena occur because of more kinetic mechanical collisions produced by milling time, milling speed, ball-to-powder ratio (BPR), and milling media (vials and balls). ImageJ software was used to measure the particle size of each sample from the different HRSEM images. The average particle size of HEA1, HEA2, and HEA3 was around $38.47 \pm 2.56$ (Figure 4a), $24.56 \pm 1.85$ (Figure 4c), and $39.45 \pm 2.1$ μm (Figure 4e), respectively. These results indicate that more fracturing mechanism was dominant in HEA2 owing to the incorporation of V atoms (increasing the entropy effect) compared to HEA1.

However, with the incorporation of Zn atoms, the observed powder particle size increased indicating the domination of cold-welding. The introduced Zn atoms were expected to dissolve more structural refinement and agglomeration of particles was expected to occur (Table 1). As a result, more peak broadening occurred in the HEA3 sample than in the other samples (Table 1 and Figure 2). The high-magnification in Figure 4b,d,e shows the formation of multi-phase solid solutions which were expected to produce improved mechanical properties. Multi-phase solid solution formation at lower magnification was confirmed by the HRSEM EDAX elemental map of all the samples which is illustrated in Figure 5. From Figure 5, the homogeneous distribution and solid solution formation were confirmed. The EDAX spectrum was used to check the chemical composition and the results are given in Table 2. The EDAX spectrum was obtained in three different places and the average was used to verify the elemental composition as shown in Table 1. The chemical composition measured by EDS was well correlated with the theoretical composition (Table 1) representing the absence of any intermetallic compounds and no segregation of any elements [36]. However, based on the gas analyzer (LECO CS 744 and ONH 836), a small amount of O, C, and H atoms due to wet milling medium (toluene), a negligible quantity of N due to contact with air during material transfer, and a negligible quantity of S due to impurities from the as-received powders were observed. HRSEM with EDS elemental mapping at higher magnification (a small area of a particle was scanned, Figure 6) was performed to confirm the multiple solid solutions and distribution of each element. The corresponding covered area spectra were also obtained. Based on the results shown in Figure 6, it is obvious that the attainment of multiple phase solid solutions has a uniform distribution of incorporated metallic elements. The formation of nanostructures and various phases of BCC, FCC, and HCP was examined using HRTEM (Figure 7). Figure 7a,c,e

show the bright-field images (BFIs) of HEA1, HEA2, and HEA3, respectively. The presence of BCC, FCC, and HCP was confirmed in the BFIs. The crystallite size was measured from several BFIs of each alloy using ImageJ software (the encircled areas in Figure 7a represent the crystallites). The average crystallite sizes of FeCrCuMnTi were $19.5 \pm 2.1$, $8.6 \pm 3.4$, and $29.7 \pm 1.8$ nm for the BCC, FCC, and HCP phases, respectively. FeCrCuMnTiV HEA produced average crystallite sizes of $22.7 \pm 3.8$, $11.1 \pm 1.5$, and $42.4 \pm 4.7$ nm for BCC, FCC, and HCP phases, respectively. The crystallite sizes of the BCC, FCC, and HCP phases of FeCrCuMnTiVZn HEA were $13.7 \pm 2.5$, $7.8 \pm 1.4$, and $20.5 \pm 3.7$ nm, respectively. The crystallite size measured from the HRTEM BFIs correlated well with the XRD results as shown in Table 1. The observed crystal structures (BCC, FCC, and HCP) were also confirmed by selective area diffraction (SAED) patterns which are shown in Figure 7b,d,f for FeCrCuMnTi, FeCrCuMnTiV, and FeCrCuMnTiVZn alloys, respectively. Further, SAED patterns of all samples produced continuous rings with halo patterns confirming the nanostructured formation as more nanocrystallites were covered in the SAED patterns.

*3.3. Examination of Powder Particle Size, Distribution, Apparent Density, Tap Density, and True Density*

The exact powder particle size and particle size distribution were examined using a particle size analyzer (Figure 8 and Table 3). Figure 8 shows the powder particle size distribution with fraction and cumulative distribution. The powder particle sizes of FeCrCuMnTi HEA at D10, D50, D90, and Davg were 24.5, 37, 53.9, and 40.79 μm, respectively. The powder particle sizes of FeCrCuMnTiV HEA were 16, 23.9, 37.9, and 26.78 μm at D10, D50, D90, and Davg. FeCrCuMnTiVZn HEA produced powder particles with sizes of 24.6, 37.7, 54.7, and 41.39 mm at D10, D50, D90, and Davg. These results demonstrate that HEA2 exhibits a smaller particle size compared to other alloys (HEA1 and HEA3) owing to more fracturing mechanisms. HEA1 and HEA3 produced a broader powder particle size distribution due to the presence of large agglomerated particles (Figure 4). These results were expected to be due to the domination of cold-welding phenomena for the same 25 h MA. A narrow powder particle size was obtained for HEA2. However, all the powder particles exhibited a uniform size distribution which is an important characteristic required for further processing. The surface area of all the developed alloys was also measured using the same particle size analyzer which were 16.35, 24.75, and 16.14 m$^2$/kg (Table 3) for HEA1, HEA2, and HEA3, respectively. The value of the surface area of the powder particles depends on the particle size and hence, HEA2 has a higher surface area value compared to other powders (HEA1, and HEA3).

The apparent density, tap density, and true density of developed HEAs are listed in Table 3. The results revealed that all the densities decreased significantly with increasing high-configurational entropy effects indicating that more structural refinement occurred in HEA3 due to more dissolution of Zn atoms (more FCC phase) leading to a smaller crystallite size and more lattice strain (Table 2 and Figure 2). Furthermore, the stress recovered powder samples produced more density values (both apparent, and tap) compared to the as-milled powder samples because of the elimination of stored strain (obtained from severe plastic deformation of powder particles during 25 h MA) in the powders after 150 °C for 1 h. These results demonstrate that the stress-recovered powders are expected to have a higher green density value after uniaxial compaction. It was noted that the true density values obtained using a pycnometer were close to the theoretical density.

**Table 2.** Chemical composition in at. wt.% of three developed HEAs taken in several regions by HRSEM-EDS.

| Alloy Code | Equiatomic Composition | Fe | Cr | Cu | Mn | Ti | V | Zn |
|---|---|---|---|---|---|---|---|---|
| HEA1 | FeCrCuMnTi | $20.75 \pm 0.06$ | $19.95 \pm 0.22$ | $19.61 \pm 0.20$ | $19.95 \pm 0.05$ | $19.79 \pm 0.06$ | - | - |
| HEA2 | FeCrCuMnTiV | $20.47 \pm 0.29$ | $16.52 \pm 0.41$ | $16.23 \pm 0.65$ | $15.89 \pm 0.04$ | $14.94 \pm 0.15$ | $15.95 \pm 0.42$ | |
| HEA3 | FeCrCuMnTiVZn | $14.27 \pm 0.11$ | $15.16 \pm 0.39$ | $13.02 \pm 0.38$ | $14.19 \pm 0.01$ | $14.59 \pm 0.22$ | $15.59 \pm 0.50$ | $13.16 \pm 0.60$ |

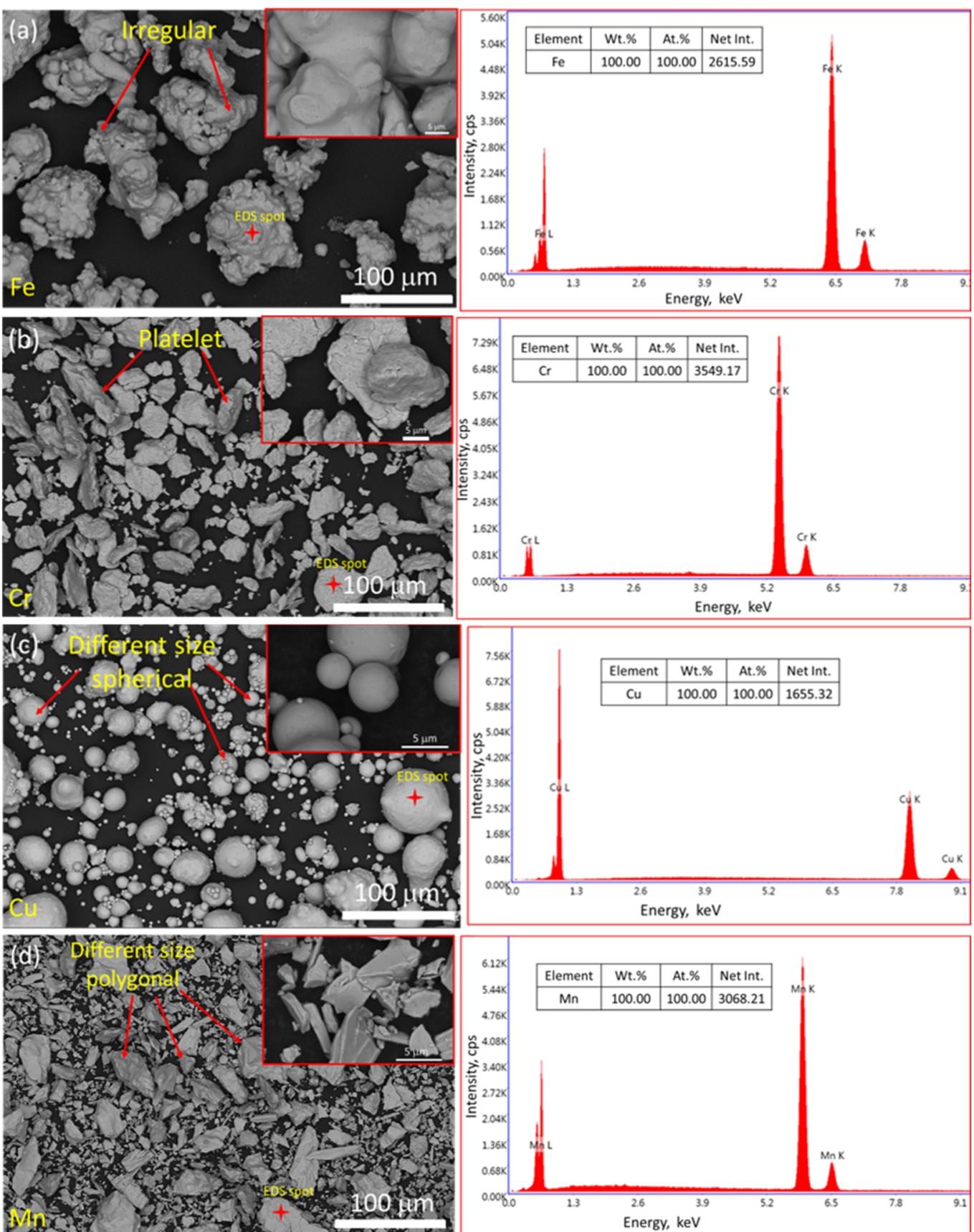

**Figure 3.** *Cont.*

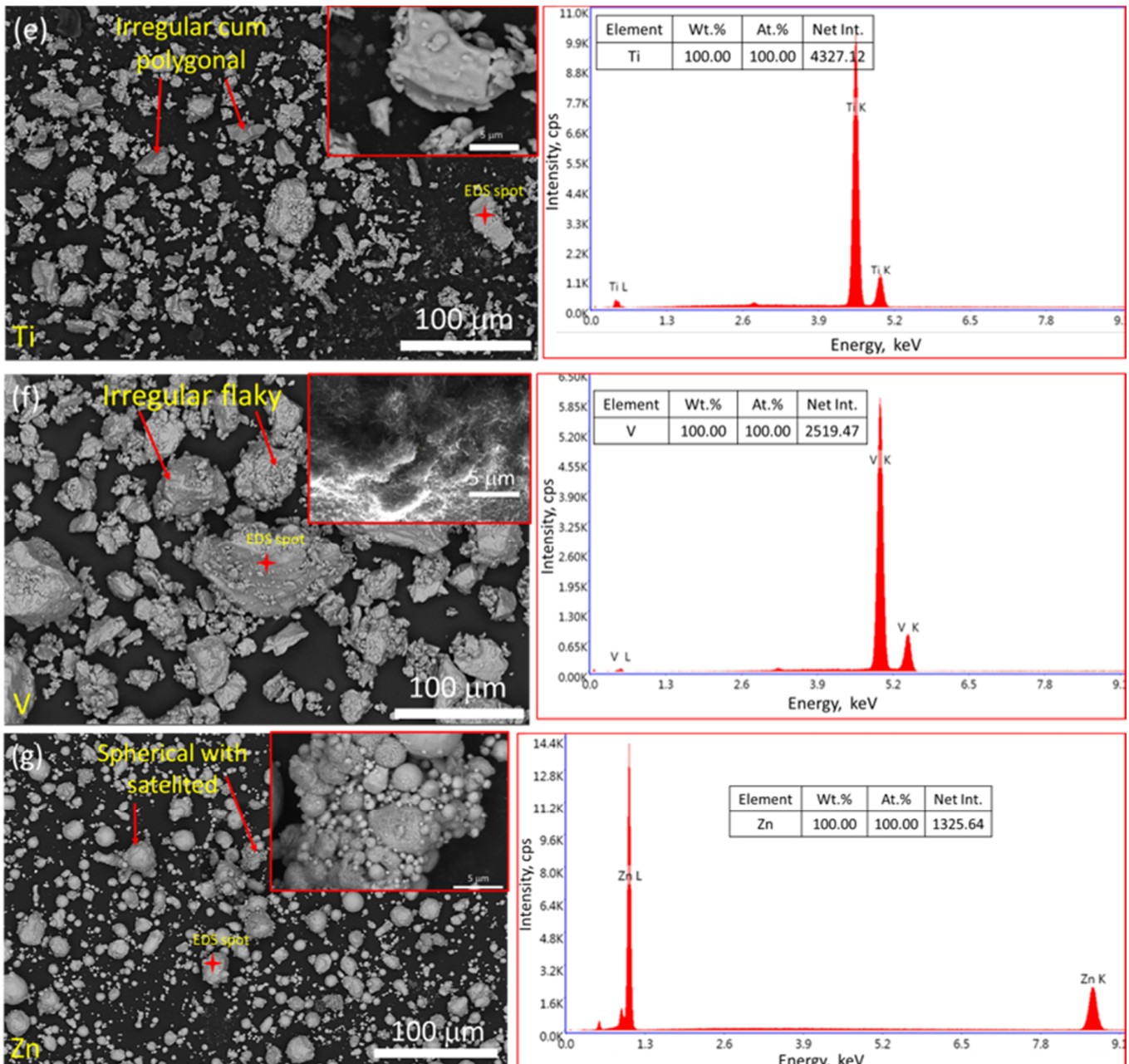

**Figure 3.** HRSEM powder surface morphology of as-received elemental powders: (**a**) Fe-irregular shape; (**b**) Cr-Platelet shape; (**c**) Cu-different size spherical shape; (**d**) Mn-different size polygonal shape; (**e**) Ti-irregular cum polygonal shape; (**f**) V-irregular; (**g**) Zn-spherical with satellite. Top right inset images describing the morphology of as-received particles at higher magnification. Right side of each image illustrating the EDS spot analyses for checking the purity of as-received elemental powders.

**Figure 4.** HRSEM BSE images over the powder surface morphology of three developed high entropy alloy powders: (**a**,**b**) FeCrCuMnTi; (**c**,**d**) FeCrCuMnTiV; and (**e**,**f**) FeCrCuMnTiVZn.

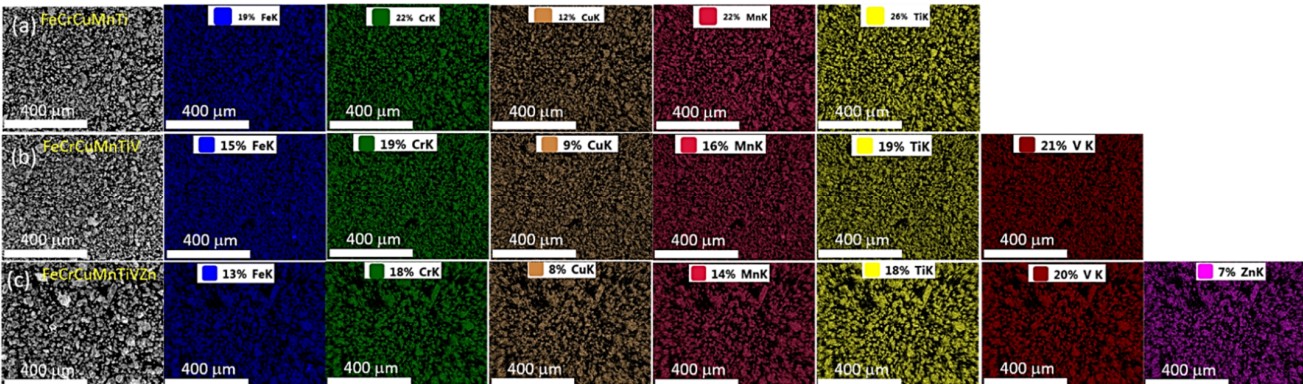

**Figure 5.** HRSEM elemental overlay map of three developed high entropy alloys: (**a**) FeCrCuMnTi; (**b**) FeCrCuMnTiV; and (**c**) FeCrCuMnTiVZn.

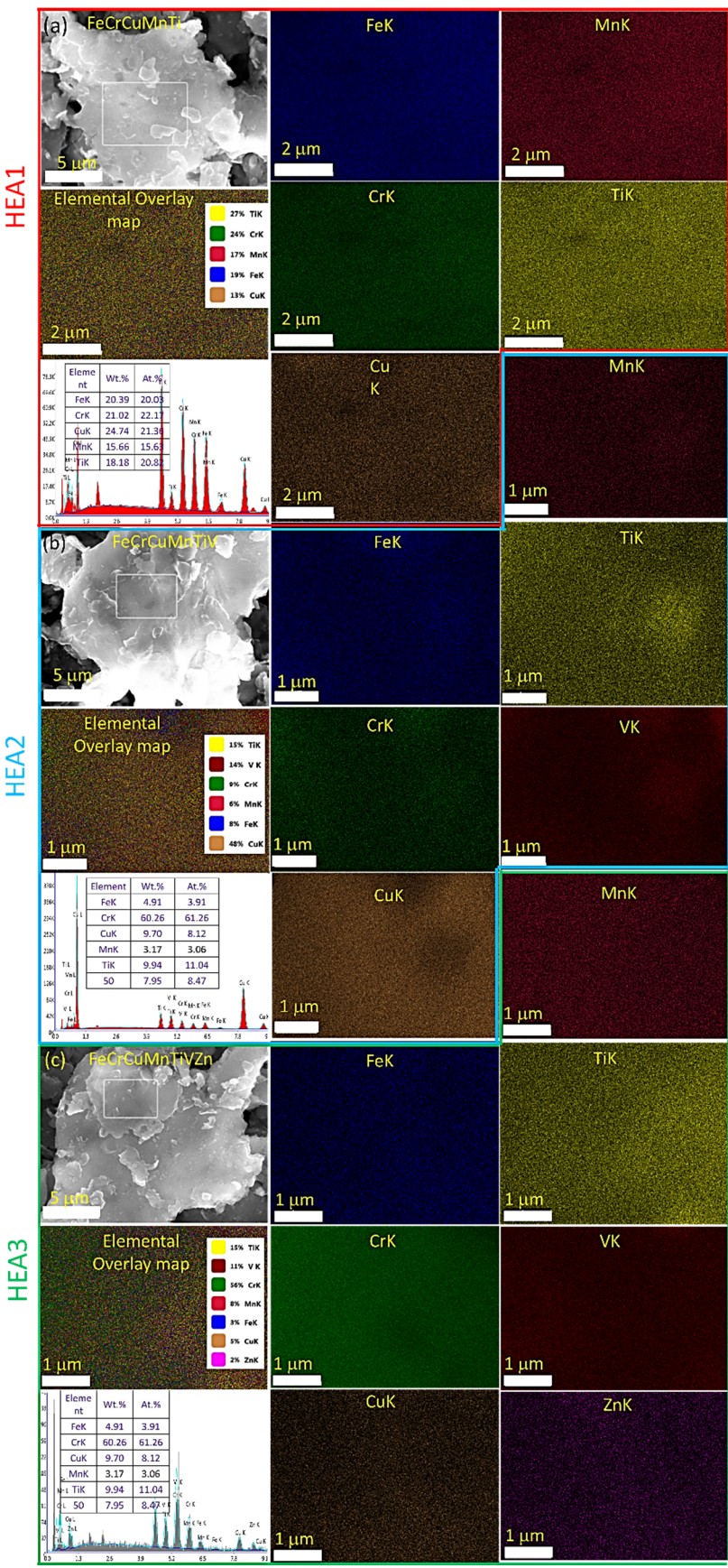

**Figure 6.** HRSEM EDS elemental mapping at high magnification showing multiple solid solutions and elemental analyses of three high-entropy alloys: (**a**) FeCrCuMnTi; (**b**) FeCrCuMnTiV; and (**c**) FeCrCuMnTiVZn.

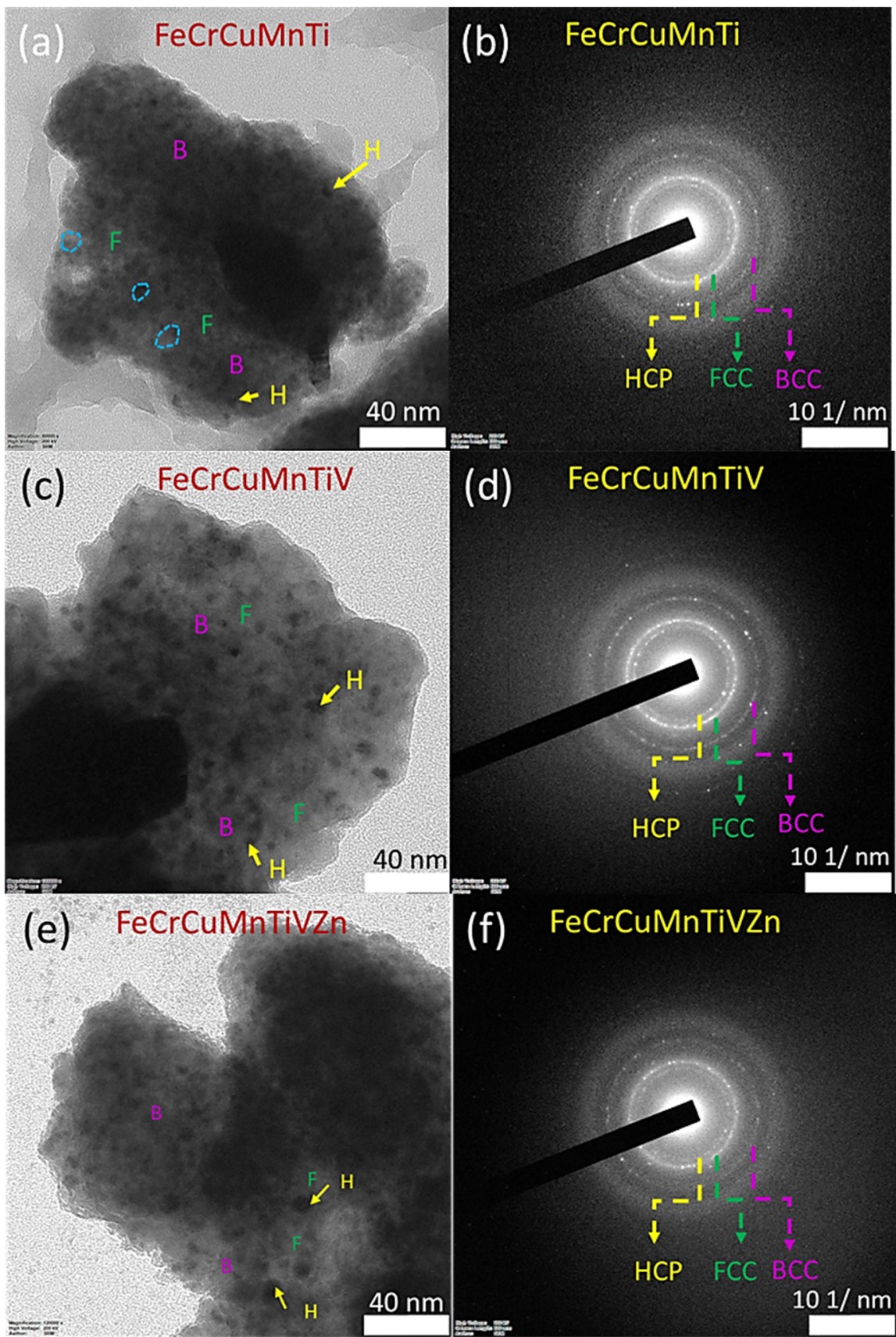

**Figure 7.** HRTEM microstructure of three developed HEAs: (**a**,**b**) FeCrCuMnTi; (**c**,**d**) FeCrCuMnTiV; (**e**,**f**) FeCrCuMnTiVZn. (**a**,**c**,**e**) Dark field images, DFI. (**b**,**d**,**f**) SAED patterns.

**Table 3.** Powder particle size, surface area, apparent density, tap density, packing at apparent density, and packing at a tapped density of three developed high entropy alloy powders.

| High Entropy Alloy Powders | Powder Particle Size | | | | Surface Area | Apparent Density | | Tap Density | | True Density | Packing at Apparent Density | | Packing at Tapped Density (150 No. of Taps) | |
|---|---|---|---|---|---|---|---|---|---|---|---|---|---|---|
| | $D_{10}$ (µm) | $D_{50}$ (µm) | $D_{90}$ (µm) | $D_{avg}$ (µm) | m²/kg | g/cm³ | | g/cm³ | | g/cm³ | % | | % | |
| | | | | | | As-Milled | Stress Recovered | As-Milled | Stress Recovered | | As-Milled | Stress Recovered | As-Milled | Stress Recovered |
| FeCrCuMnTi | 24.5 | 37 | 53.9 | 40.79 ± 1.84 | 16.35 | 2.60 ± 0.045 | 2.73 ± 0.053 | 3.30 ± 0.032 | 3.47 ± 0.028 | 6.22 ± 0.011 | 41.69 ± 0.563 | 43.68 ± 0.489 | 52.88 ± 0.657 | 55.52 ± 0.258 |
| FeCrCuMnTiV | 16 | 23.9 | 37.9 | 26.78 ± 2.89 | 24.75 | 2.62 ± 0.015 | 2.75 ± 0.023 | 3.40 ± 0.013 | 3.58 ± 0.031 | 6.52 ± 0.018 | 39.98 ± 0.687 | 41.92 ± 0.202 | 51.96 ± 0.326 | 54.57 ± 0.272 |
| FeCrCuMnTiVZn | 24.6 | 37.7 | 54.7 | 41.39 ± 1.20 | 16.14 | 2.49 ± 0.017 | 2.62 ± 0.032 | 3.08 ± 0.012 | 3.24 ± 0.024 | 6.64 ± 0.014 | 37.47 ± 0.723 | 39.39 ± 0.277 | 46.40 ± 0.468 | 49.08 ± 0.240 |

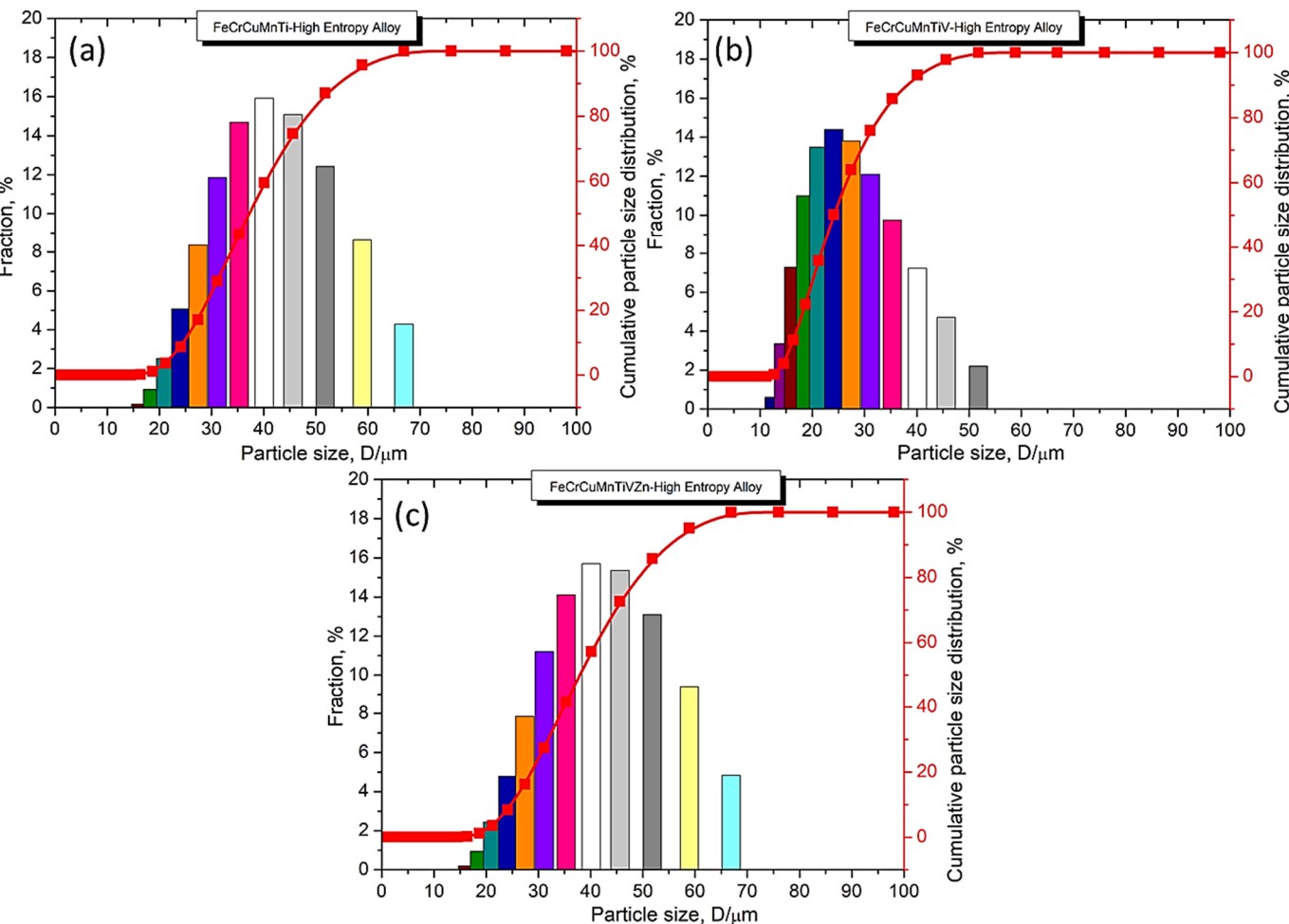

**Figure 8.** Particle size distribution of three developed high entropy alloy powders: (**a**) HEA1-FeCrCuMnTi; (**b**) HEA2-FeCrCuMnTiV; and (**c**) HEA3-FeCrCuMnTiVZn.

### 3.4. Densification Behavior of FeCrCuMnTi, FeCrCuMnTiV, and FeCrCuMnTiVZn HEAs

The densification behavior of the three developed equiatomic HEAs under as-milled and milled under stress recovery conditions was investigated to explore the powder particle packing performance during its consolidation. The densification behavior was experimentally studied with respect to the applied pressure. Five linear compaction equations and three non-linear compaction equations were applied to the experimental data to determine the best model for predicting the relative density [37]. For instance, the theoretical density, green density, and the relative density of green pellets at 1100 MPa are listed in Table 4. The results demonstrated that HEA2 exhibited a higher value of relative density in both conditions compared to other samples (HEA1 and HEA3) because of the greater particle size reduction, less structural refinement, and domination of fracturing phenomena in addition to high-configurational entropy, cocktail effect, and sluggish effect. HEA3 produced a lower value of relative density in both conditions due to more structural refinement, presence of more lattice strain, formation of more FCC refined phases, and a lower amount of HCP and BCC phases. The stress-recovered powder samples produced an improved relative density compared to the as-milled powder samples. For example, the HEA2 sample exhibited a relative density of 0.8284, which was approximately 11.105% higher than that of the as-milled HEA2 sample. These results demonstrated that the elimination of internal stresses stored and lattice strain during long 25 h MA in the powder samples was expected to occur after stress recovery at 150 °C for 1 h. Because of this, the powders were

offered a lower amount of resistance during packing/compaction leading to improved densification. The HRSEM secondary electron (SE) microstructures with EDS elemental overlay maps of the stress-recovered green pellets compacted at 1100 MPa are shown in Figure 9. The HRSEM-SE microstructure was taken at two different magnifications (low and high). The presence of BCC, FCC, and HCP phases with porosity was observed in all the stress-recovered samples. From the high-magnification images in Figure 9 (right side), it was observed that more porosity was observed in HEA1 and HEA3 samples compared to HEA2. This was attributed to less structural refinement, and a large particle size reduction in the HEA2 sample. Figure 9c,f,i shows the HRSEM EDS elemental overlay maps of the green compacts after stress recovery. These results demonstrate a homogeneous structure with an effective solid solution of incorporated elements in the developed alloys.

The compressibility/densification performance of FeCrCuMnTi, FeCrCuMnTiV, and FeCrCuMnTiVZn HEAs are shown in Figure 10a,b for no stress recovery and stress recovered powders respectively on the same scale of the x- and y-axis. Increasing the compaction pressure increased the relative density. The densification curves in both conditions exhibit three stages: powder particle rearrangement, plastic deformation, and powder particle impingement. The stress recovered powder samples reached the highest relative density early compared to the sample without stress recovery. The elimination of internal stresses in the powders accelerated all three stages. During the first stage (powder particle rearrangement), the as-milled powder samples had a compaction pressure range of 1100 MPa, whereas the stress recovered samples exhibited a compaction pressure range of 0–95 MPa which was almost the same. For the as-milled condition, the relative densities obtained at the end of this stage were approximately 0.478, 0.518, and 0.4665 for HEA1, HEA2, and HEA3, respectively. The relative densities of the stress-recovered conditions were approximately 0.510, 0.528, and 0.480 for HEA1, HEA2, and HEA3, respectively. However, during the second stage (plastic deformation), 110–510 MPa and 95–275 MPa were the compaction pressure ranges for the as-milled powder samples and stress recovered samples respectively. At the end of this stage, the densities of the as-milled powders were approximately 0.508, 0.525, and 0.478 for HEA1, HEA2, and HEA3, respectively. The relative densities of the stress-recovered powders were 0.573, 0.591, and 0.547 for the HEA1, HEA2, and HEA3, respectively. Similarly, during particle impingement (third stage), the powder compaction ranges for as-milled samples and stress-recovered samples were 510–1100 MPa and 275–1100 MPa, respectively. At the end of this stage, the relative densities of the as-milled powders were 0.685, 0.745, and 0.676 for HEA1, HEA2, and HEA3, respectively. The relative densities of the stress-recovered powders were 0.788, 0.828, and 0.754, respectively. These results revealed that stress-relieving accelerated the densification, and consequently the stress-relieved samples exhibited a higher relative density. In addition, FeCrCuMnTiV HEA produced a higher relative density compared to other samples due to greater particle size reduction, less structural refinement, and a lower amount of dislocation density, high-configurational entropy, and a lower amount of sluggish and cocktail effects.

The densification behavior of all the developed alloys under both conditions was investigated using various linear and non-linear empirical models. These models provide a relationship between compaction pressure and relative density. The relative densities of all samples can be easily predicted using these theoretical models based on the experimental results. Five linear models (Balshin's [38], Heckel's [39], Ge's [40], Panelli and Ambrosio Filho's [41], Kawakita's [42]) and three non-linear models (Shapiro's [43], Cooper and Eaton's [44], and Van Der Zwan and Siskens [45]) were used in this study.

Linear models (Equations (6)–(10)):

Balshin's [38]:

$$\frac{1}{D_R} = K \ln P + A \tag{6}$$

Heckel's [39]:

$$\ln\left(\frac{1}{1 - D_R}\right) = KP + A \tag{7}$$

**Table 4.** Theoretical density, green density, and relative density of three developed HEAs measured at 1100 MPa.

| Alloy Composition | Theoretical Density (g/cm$^3$) | Green Density (g/cm$^3$) | | Relative Density (%) | |
|---|---|---|---|---|---|
| | | No Stress Recovery | Stress Recovered | No Stress Recovery | Stress Recovered |
| FeCrCuMnTi | 6.25 | 4.29 | 4.93 | 68.56 | 78.85 |
| FeCrCuMnTiV | 6.56 | 4.89 | 5.43 | 74.56 | 82.84 |
| FeCrCuMnTiVZn | 6.65 | 4.50 | 5.01 | 67.65 | 75.45 |

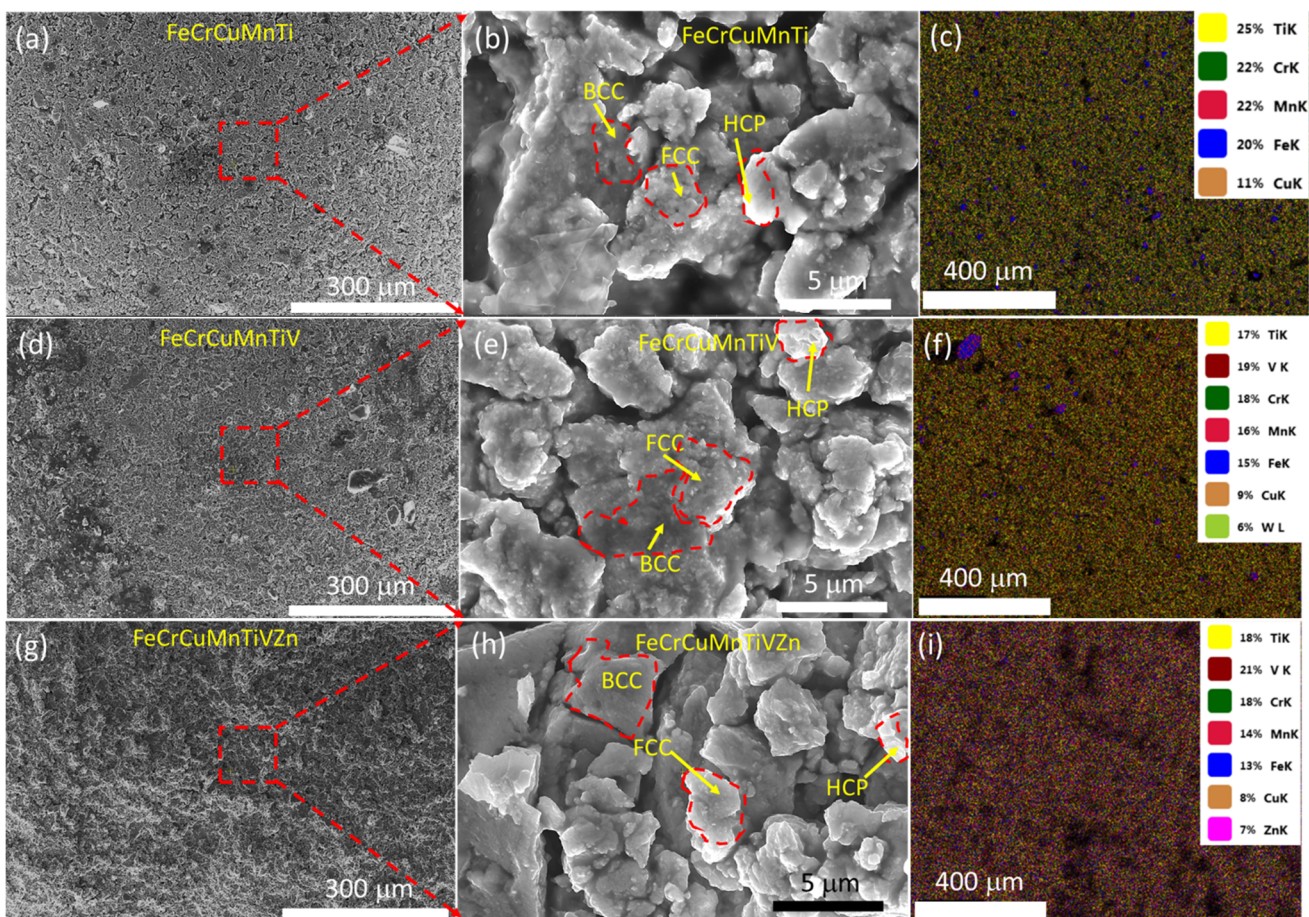

**Figure 9.** HRSEM SE microstructures and EDS overlay map of green compacts pressed at 1100 MPa of three developed HEAs after stress recovery: (**a–c**) FeCrCuMnTi; (**d–f**) FeCrCuMnTiV; and (**g–i**) FeCrCuMnTiVZn.

Ge's [40]:

$$\log\left(\ln\frac{1}{1 - D_R}\right) = K\log P + A \tag{8}$$

Panelli and Ambrosio Filho's [41]:

$$\ln\left(\frac{1}{1 - D_R}\right) = K\sqrt{P} + A \tag{9}$$

Kawakita's [42]:

$$\frac{D_R}{D_R - D_o} = \frac{K}{P} + A \tag{10}$$

Non-linear models (Equations (11)–(13)):
Shapiro's [43]:

$$\ln(1 - D_R) = \ln(1 - D_o) - CP - B\sqrt{P} + A \tag{11}$$

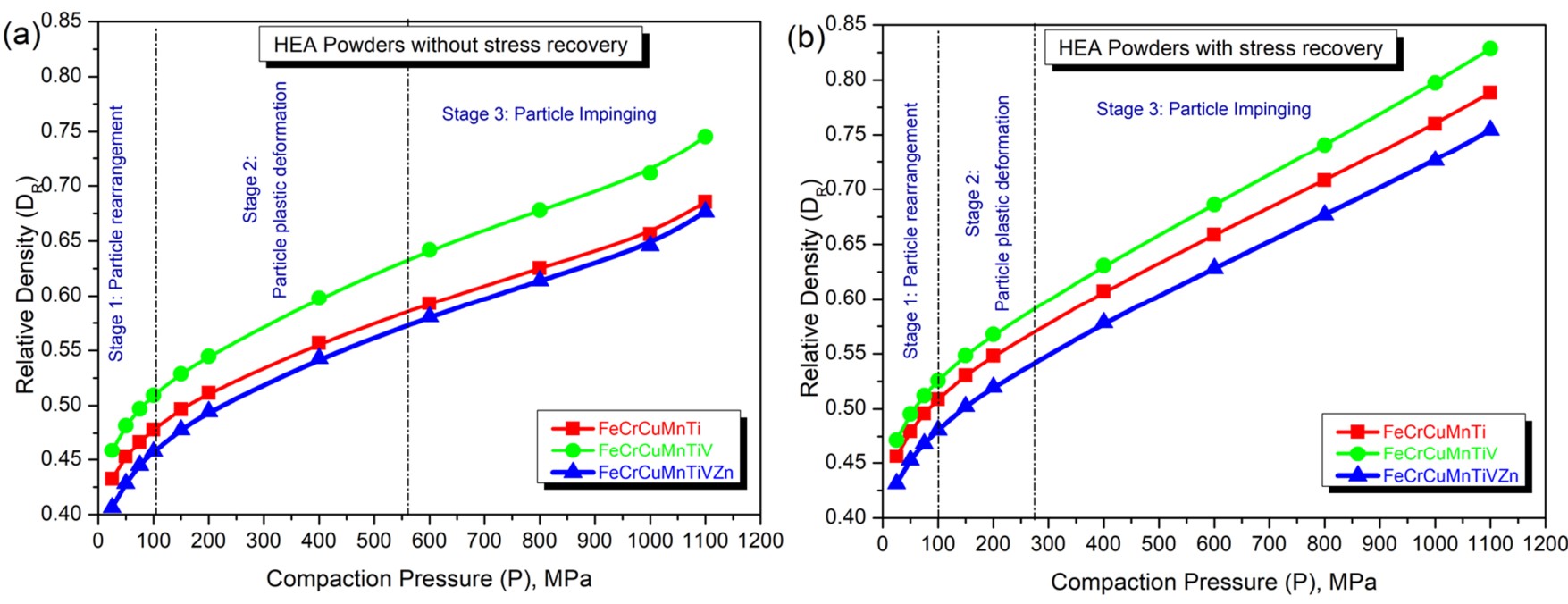

**Figure 10.** Compaction behavior of three developed high entropy alloy powders of FeCrCuMnTi, FeCrCuMnTiV, and FeCrCuMnTiVZn with different conditions: (**a**) without stress recovery and (**b**) with stress recovery.

Cooper and Eaton's [44]:

$$\frac{D_R - D_o}{D_R(1 - D_o)} = a_1 \exp\left(\frac{-k_1}{P}\right) + a_2 \exp\left(\frac{-k_2}{P}\right) \qquad (12)$$

Van Der Zwan and Sisken's [45]:

$$\frac{D_R - D_o}{(1 - D_o)} = a \ \exp\left(\frac{-k}{P}\right) \qquad (13)$$

where P is the applied compaction pressure in MPa, and $D_R$ is the relative density. Other symbols (A, a, $a_1$, $a_2$, K, $k_1$, and $k_2$) used in these models are parameters that depend on each model. The numeric value of each parameter was determined by curve fitting technique and these parameters can help us to investigate and predict the densification of the powder samples. Figures 11 and 12 show the densification behavior of the three developed HEAs fitted by linear and non-linear models, respectively. Table 5 lists the coefficients and intercepts corresponding to each model. Based on Table 5 and Figure 11, Balshin's Equation (6) and Heckel's Equation (7) are well-fitted linear models as these two models exhibit a regression coefficient of more than 0.97. Whereas Shapiro's Equation (11) and Cooper and Eaton's Equation (12) have produced regression coefficients greater than 0.99 indicating the good accuracy of non-linear models compared to linear models. Hence, Balshin's Equation (6), Heckel's Equation (7), Shapiro's Equation (11), and Cooper and Eaton's Equation (12) can be used to predict the densification behavior of FeCrCuMnTi, FeCrCuMnTiV, and FeCrCuMnTiVZn HEAs.

**Table 5.** Various parameters associated with the linear and non-linear compaction equations obtained for quinary thermoelectric compacts.

| Compaction Equation | Parameter | FeCrCuMnTi (HEA1) | | FeCrCuMnTiV (HEA2) | | FeCrCuMnTiVZn (HEA3) | |
|---|---|---|---|---|---|---|---|
| | | WOS | WS | WOS | WS | WOS | WS |
| Balshin Equation (6) | A | 3.09845 | 3.06949 | 2.95709 | 2.99541 | 3.34952 | 3.27144 |
| | K | −0.22402 | −0.24659 | −0.22102 | −0.24461 | −0.25754 | −0.26668 |
| | $R^2$ | 0.97874 | 0.97134 | 0.97807 | 0.96946 | 0.98668 | 0.97168 |
| Heckel's Equation (7) | A | 0.59261 | 0.61384 | 0.64873 | 0.64299 | 0.5524 | 0.57058 |
| | K | 0.00050 | 0.00081 | 0.00062 | 0.00095 | 0.00051 | 0.00073 |
| | $R^2$ | 0.98885 | 0.99439 | 0.98671 | 0.98705 | 0.98654 | 0.99573 |
| Ge's Equation (8) | A | −0.54121 | −0.61291 | −0.54458 | −0.62799 | −0.59842 | −0.63899 |
| | K | 0.18382 | 0.24233 | 0.20618 | 0.26160 | 0.19908 | 0.23788 |
| | $R^2$ | 0.93834 | 0.90832 | 0.93080 | 0.89714 | 0.94849 | 0.91319 |
| Panelli and Ambrosio Filho Equation (9) | A | 0.4522 | 0.39226 | 0.45956 | 0.37035 | 0.40766 | 0.37123 |
| | K | 0.01950 | 0.03141 | 0.02485 | 0.03703 | 0.02003 | 0.02817 |
| | $R^2$ | 0.98261 | 0.95769 | 0.97648 | 0.94458 | 0.98531 | 0.96354 |
| Kawakita's Equation (10) | A | 2.71834 | 2.4562 | 2.54322 | 2.40788 | 2.54003 | 2.48555 |
| | K | 135.8672 | 120.53758 | 107.0222 | 112.97297 | 185.8354 | 146.22189 |
| | $R^2$ | 0.90689 | 0.88789 | 0.88034 | 0.8753 | 0.96324 | 0.9095 |
| Shapiro Equation (11) | A | −0.81222 | −0.90605 | −0.89129 | −0.98031 | −0.74443 | −0.8176 |
| | B | $−6.89 \times 10^{-4}$ | $−6.468 \times 10^{-4}$ | $−8.507 \times 10^{-4}$ | $−4.75 \times 10^{-4}$ | $−6.970 \times 10^{-4}$ | $−6.342 \times 10^{-4}$ |
| | C | $−1.55 \times 10^{-7}$ | $−1.013 \times 10^{-6}$ | $−3.736 \times 10^{-7}$ | $−1.663 \times 10^{-6}$ | $−1.342 \times 10^{-7}$ | $−7.2824 \times 10^{-7}$ |
| | $R^2$ | 0.99059 | 0.99532 | 0.99021 | 0.99386 | 0.99002 | 0.99608 |
| Cooper and Eaton's Equation (12) | $a_1$ | 0.32711 | 0.34005 | 0.4907 | 0.57359 | 0.49666 | 0.5512 |
| | $a_2$ | 0.49837 | 0.55277 | 0.3198 | 0.34613 | 0.38257 | 0.34836 |
| | $k_1$ | 79.0513 | 72.8862 | 826.446 | 800.000 | 961.5384 | 787.4015 |
| | $k_2$ | 961.5384 | 793.6507 | 71.9424 | 74.96251 | 77.16049 | 76.2195 |
| | $R^2$ | 0.9992 | 0.9994 | 0.99806 | 0.99938 | 0.9993 | 0.99946 |
| Van Der Zwan and Sisken's Equation (13) | a | 0.46183 | 0.71326 | 0.54438 | 0.79743 | 0.46777 | 0.67931 |
| | k | 384.6153 | 505.0505 | 401.6064 | 526.3157 | 348.4320 | 500.000 |
| | $R^2$ | 0.95966 | 0.96511 | 0.95371 | 0.96607 | 0.95918 | 0.96667 |

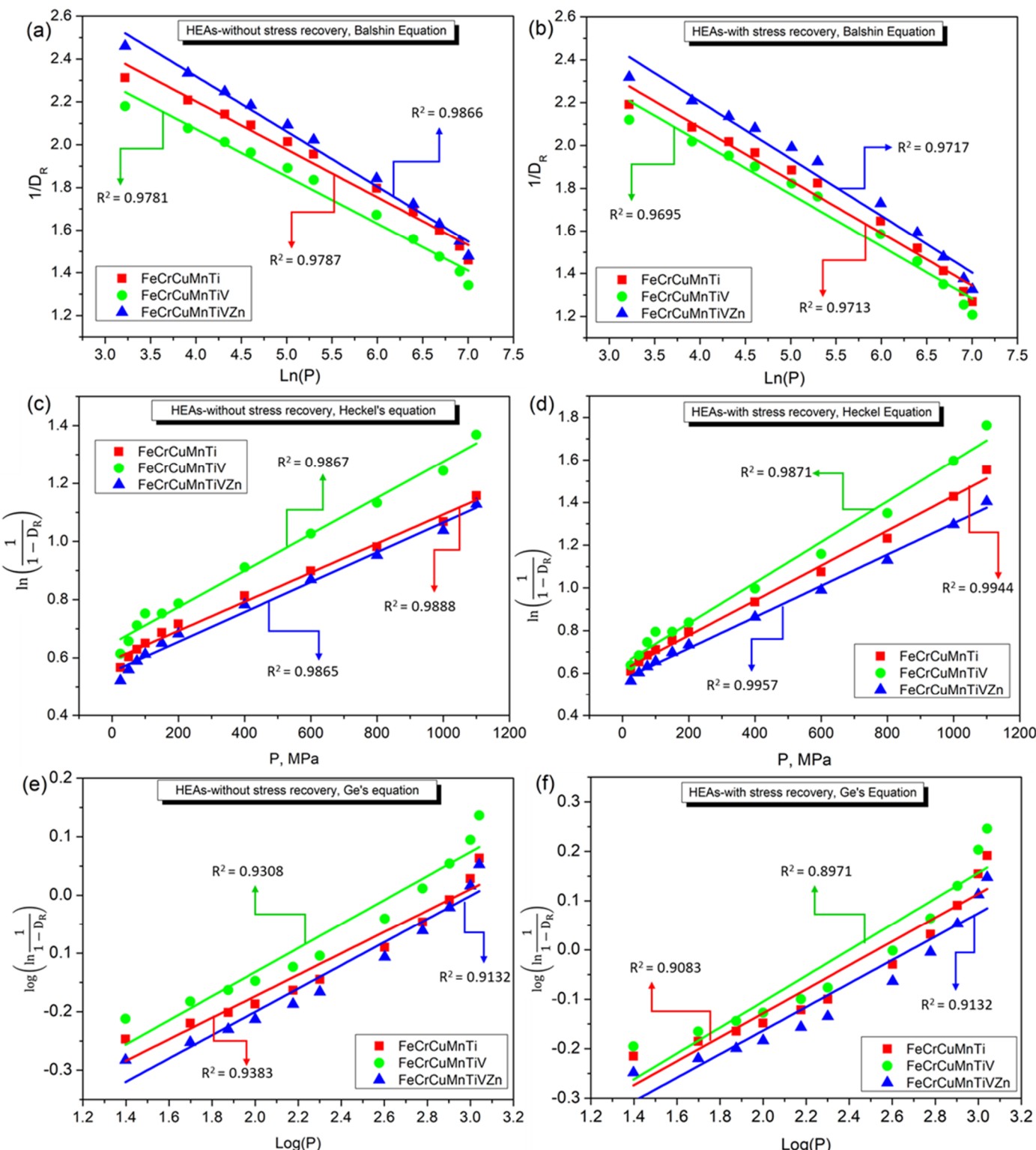

**Figure 11.** *Cont.*

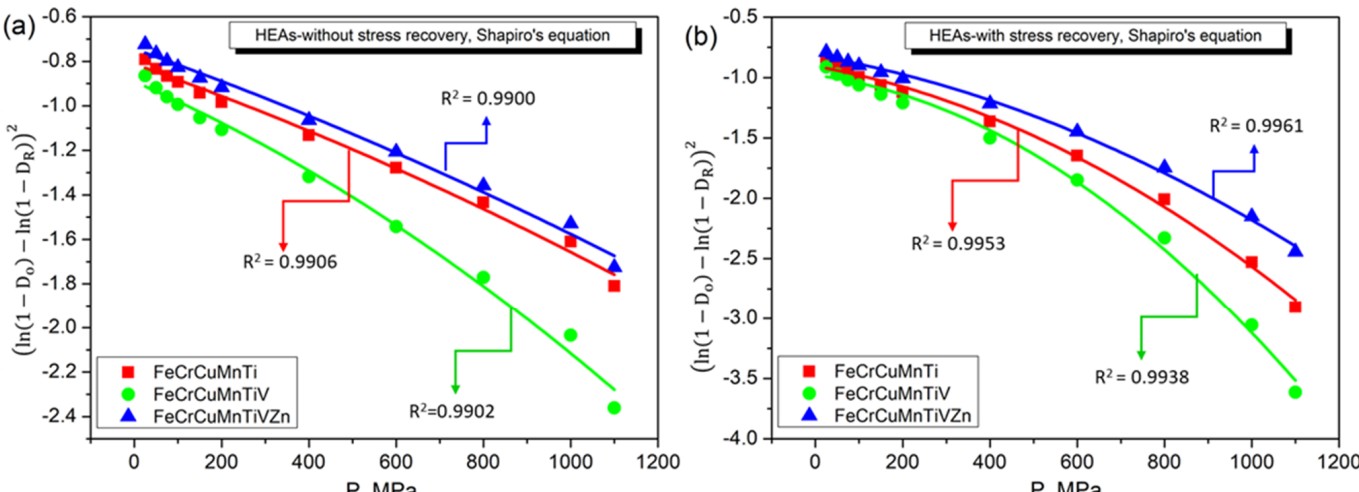

**Figure 11.** Densification behavior fitted by various linear models for the three developed nanostructured HEAs for as-milled and stress recovered conditions: (**a**,**b**) Balshin's Equation (6); (**c**,**d**) Heckel's Equation (7); (**e**,**f**) Ge's Equation (8); (**g**,**h**) Panelli and Ambrosio Filho's Equation (9); (**i**,**j**) Kawakita's Equation (10).

**Figure 12.** *Cont*.

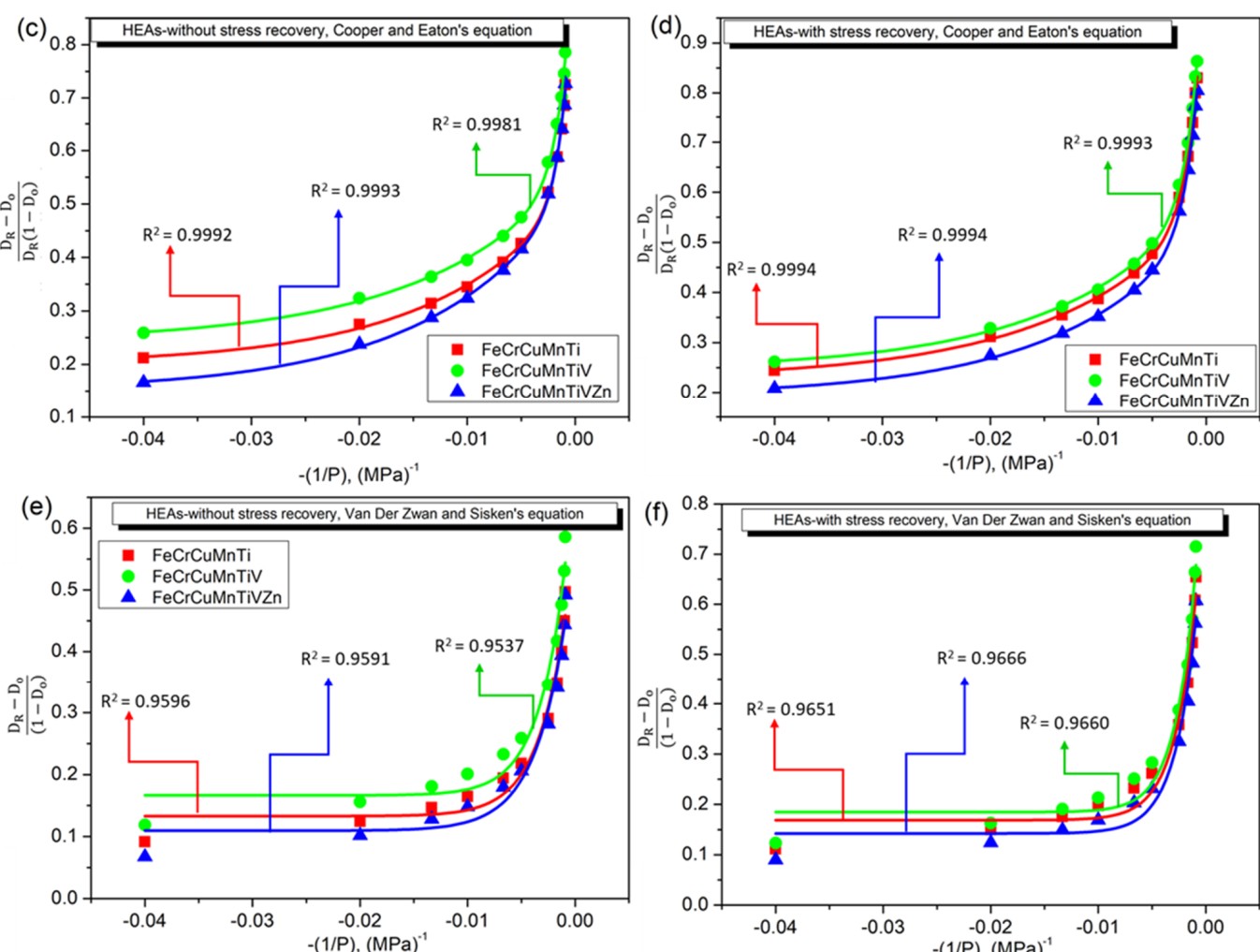

**Figure 12.** Densification behavior fitted by various non-linear models for the three developed nanostructured HEAs for as-milled and stress recovered conditions: (**a**,**b**) Shapiro's Equation (11); (**c**,**d**) Cooper and Eaton's Equation (12); (**e**,**f**) Van Der Zwan and Sisken's Equation (13).

## 4. Discussion

Three nanostructured HEAs, namely, FeCrCuMnTi (HEA1), FeCrCuMnTiV (HEA2), and FeCrCuMnTiVZn (HEA3) were synthesized by MA at 25 h in this study. The solid solution formation of each alloy was confirmed by XRD patterns (Figure 2) and the higher magnification of the HRSEM images (Figures 4 and 6) represents the presence of BCC, FCC, and HCP phases. FCC was found to be the major phase in each alloy. The densification behavior of the developed HEAs was highly influenced by the incorporated elements, solid solution formation, amount of structural refinements (crystallite size reduction, lattice strain, lattice parameters), and powder conditions (as-milled and stress-recovered). In addition, thermodynamic parameters (free energy, $\Delta G_{mix}$, amount of mixing entropy, $\Delta S_{mix}$, and melting point of incorporated elements), and several physio-chemical parameters (atomic radius, valence electrons, and electronegativity) were expected to influence the densification behavior and hence, these values were determined based on several equations (Equations (14)–(22)) [46–48]. The calculated values of the physio-chemical and thermodynamic parameters are illustrated in Table 6. The enthalpy of mixing ($H_{mix}^{AB}$) of each pair of elements corresponding to the design of the developed HEAs is given in Table 7 which was obtained based on Miedema's model [49].

**Table 6.** Calculated physio-chemical and thermodynamic values for the three developed nanostructured HEAs.

| Equiatomic Alloy Composition | ΔHmix, kJ/mol | Δsmix, J/mol·k | Melting Temp, °C | Ω | Atomic Radius Difference (δ), % | ΔG_mix, J/mol | VEC | Expected Phases | Actual Observed Phases | PED (Δx) |
|---|---|---|---|---|---|---|---|---|---|---|
| FeCrCuMnTi | −1.76 | 13.38 | 1488 | 13.3930 | 1.44% | −5747.50 | 7.20 | BCC, FCC | BCC, FCC, HCP | 0.15 |
| FeCrCuMnTiV | −2.00 | 14.90 | 1588 | 13.6441 | 1.70 | −6439.21 | 6.83 | BCC, FCC | BCC, FCC, HCP | 0.14 |
| FeCrCuMnTiVZn | −2.53 | 16.18 | 1396 | 10.6883 | 1.79 | −7353.93 | 7.57 | BCC, FCC | BCC, FCC, HCP | 0.13 |

**Table 7.** Chemical mixing enthalpy ($\Delta H_{ij}^{mix}$, kJ/mol) of three developed equiatomic high entropy alloys determined by Miedema's approach [34].

| Element (Atomic Size, nm) | Fe | Cr | Cu | Mn | Ti | V | Zn |
|---|---|---|---|---|---|---|---|
| Fe (0.14) | - | −1 | 13 | 0 | −17 | −7 | 4 |
| Cr (0.14) | - | - | 12 | 2 | −7 | −2 | 5 |
| Cu (0.135) | - | - | - | 4 | −9 | 5 | 1 |
| Mn (0.14) | - | - | - | - | −8 | −1 | −6 |
| Ti (0.14) | - | - | - | - | - | −2 | −15 |
| V (0.135) | - | - | - | - | - | - | −2 |
| Zn (0.135) | - | - | - | - | - | - | - |

Gibbs free energy (Equation (9)):

$$\Delta G_{mix} = \Delta H_{mix} - T\Delta S_{mix} \tag{14}$$

'*T*' indicates the temperature in *K*.
Mixing entropy (Equation (10)):

$$\Delta S_{mix} = -R \sum_{i=1}^{N} (C_i ln C_i) \tag{15}$$

'*R*' represents the gas constant in J/kgK, '*N*' represents the number of elements, and $C_i$ indicates the atomic fraction of *i*th elements.
Mixing enthalpy (Equation (11)):

$$\Delta H_{mix} = \sum_{i=1, i\neq j}^{N} \left( 4\Delta H_{mix}^{AB} C_i C_j \right) \tag{16}$$

$C_j$ denotes the atomic fraction of the *j*th elements, $H_{mix}^{AB}$ represents the enthalpy of mixing between pairs of elements according to Miedema's model [38].
Thermodynamic parameter (Ω) (Equation (12)):

$$\Omega = \left| \frac{T_m \Delta S_{mix}}{\Delta H_{mix}} \right| \tag{17}$$

$T_m$ denotes the alloy melting point.
The deviation in atomic radius (δ) (Equation (13)):

$$\delta = \sqrt{\sum_{i=1}^{N} C_i \left(1 - \frac{r_i}{\bar{r}}\right)^2} \tag{18}$$

$r_i$ represents the atomic radius of *i*th element.

The mean value of atomic radius ($\bar{r}$) (Equation (14)):

$$\bar{r} = \sum_{i=1}^{N} C_i r_i \tag{19}$$

Valence electron concentration (*VEC*) (Equation (15)):

$$VEC = \sum_{i=1}^{N} C_i (VEC)_i \tag{20}$$

The difference in the Pauling electronegativity difference (PED), $\Delta x$, (Equation (16)):

$$\Delta X = \sqrt{\sum_{i=1}^{N} C_i \left(X_i - \overline{X}\right)^2} \tag{21}$$

$X_i$ represents the Pauling electronegativity of the ith element and $\overline{X}$ denotes the mean of Pauling electronegativity (Equation (17)):

$$\overline{X} = \sum_{i=1}^{N} C_i X_i \tag{22}$$

In general, the Gibbs free energy ($\Delta G_{mix}$) should be minimized to obtain a solid solution and phase stability in HEAs [39]. Based on the calculated physio-chemical and thermodynamic values listed in Table 6, the values of $\Delta G_{mix}$ are −5747.50, −6439.21, and −7353.93 J/mol for HEA1, HEA2, and HEA3, respectively, indicating the possibility of a solid solution owing to the −ve values. The value of the thermo-dynamic parameter ($\Omega$) also represents the solid solution formation and the condition is $\Omega > 10$ [40]. Where, all three HEAs possessed more than 10, representing the highest possible solid solution. In addition, the atomic radius deviation ($\delta$) of the incorporated elements should be less than 6.67% which is another condition for solid solution formation [41]. As shown in Table 6, the values of $\delta$ are 1.44%, 1.70%, and 1.79% for HEA1, HEA2, and HEA3, respectively, indicating a greater potential for solid solution formation. The VEC value predicted the type of phase formation. If VEC > 8, a stable FCC phase and unstable BCC phase can be expected, whereas, a stable BCC phase and unstable FCC phase can be expected if VEC < 6.8. In the present work, all the alloys possessed VEC values greater than 6.8 and less than 8, indicating the possible formation of mixed BCC and FCC phases. The value of PED ($\Delta x$) defines the phase stability of HEAs with $\Delta x \geq 0.13$ [48]. Here, all the developed alloys possessed a PED greater than 0.13 (Table 6), indicating the achievement of phase stability. These results are consistent with the XRD, HRSEM, and HRTEM results (Figure 2, Figure 4, and Figure 7) except for the formation of the HCP phase. The formed HCP phase was expected to incorporate Ti elements which might have boosted HCP phase formation. The conditions based on empirical formulae are an assumption and it is not necessary to apply strictly to predict the crystal structures in HEAs. However, these empirical equations can be used to identify the possibility of solid solution formation and not for crystal structure determination [36]. The obtained results demonstrate the formation of multiple solid solutions with three crystal structures (BCC, FCC, and HCP, Figure 13). Figure 13 shows the possible mixing phases in the equiatomic FeCrCuMnTi, FeCrCuMnTiV, and FeCrCuMnTiVZn HEAs as evidenced by XRD, HRSEM, and HRTEM (Figure 2, Figure 4, Figure 6, and Figure 7). In general, the number of elements, concentrations, and types of elements define the alloy formations. The stability and type of phase formation mainly depend on the Gibbs free energy which has to be more or less negative values of $\Delta G_{mix}$. Otherwise, the possibility of intermetallic compound formation exists. Four possible reactions usually occur during the mixing of metallic elements in an equiatomic ratio, namely, a single solid solution, multiple solid solutions, intermetallic with a solid solution, and spinodal

decomposition [50–53]. The formation of single solid solutions or multiple solid solutions offers several features, such as high mechanical properties in terms of strength and ductility, a stable phase at elevated temperature, and, more wear and corrosion resistance, suitable for various applications, including, but not limited to, aerospace, structural, nuclear, space, and automotive parts. The formation of intermetallic compounds promotes their brittle characteristics. Based on the observed results for the three newly-developed nanostructured HEAs (FeCrCuMnTi, FeCrCuMnTiV, and FeCrCuMnTiVZn), all the alloys produced multiple solid solutions (Figure 6) and no intermetallic compounds were observed in any of the results. Figure 13 shows the possibility of multiple solid solutions from the mixing of equiatomic metallic elements. Several researchers have observed multiple solid solutions during the development of HEAs [54–56]. Hence, these three HEAs can be suggested for potential applications in various fields.

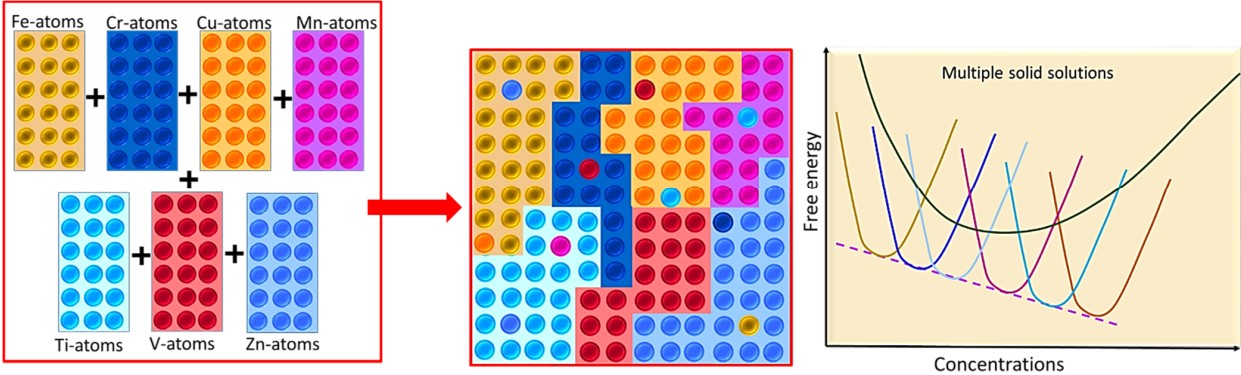

**Figure 13.** Possible mixing phases among the incorporated equiatomic metallic elements to form multiple solid solutions occurred in the three developed nanostructured HEAs.

The effect of the incorporation of V and Zn in the base FeCrCuMnTi can be explained based on the diffusion rate of the incorporated elements (Figure 14). HEA1 consists of one low melting element (Cu), two intermediate melting elements (Mn and Fe), and two higher melting elements (Cr and Ti). The medium diffusion rate was expected to occur in HEA1 and hence, HEA1 produced the average densification behavior compared to the other two alloys (Figures 10–12). However, the incorporation of the higher melting element of V in the base FeCrCuMnTi alloy termed as FeCrCuMnTiV HEA2 was expected to decrease the diffusion rate during the MA process leading to less structural refinement, and consequently increasing the densification behavior of the other two alloys (Figures 10–12). The addition of the low melting element of Zn in FeCrCuMnTiV termed as FeCrCuMnTiVZn HEA3 was expected to increase the diffusion rate during the MA process leading to more structural refinement and hence, HEA3 produced less densification behavior between the two alloys.

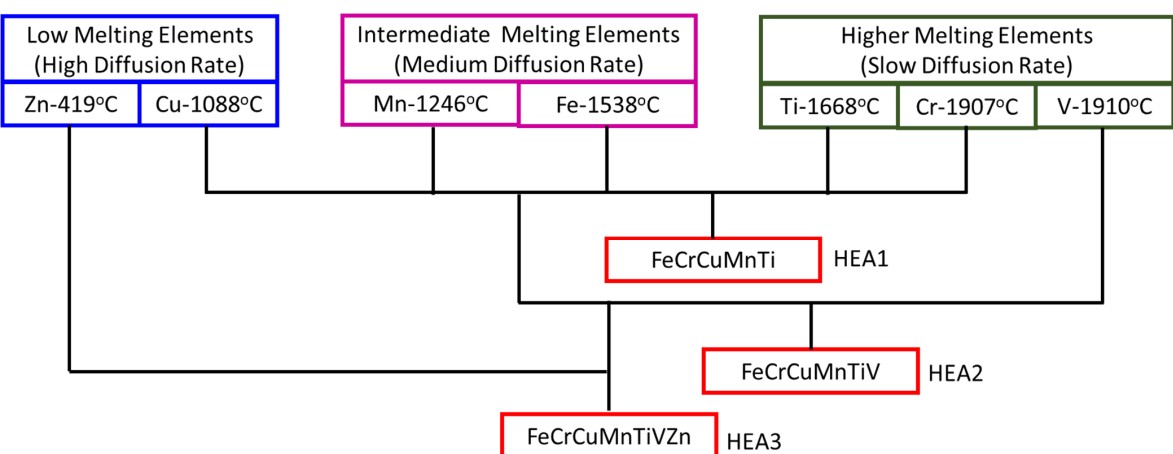

**Figure 14.** Schematic representing the diffusion rate of each HEAs defining their densification performances.

## 5. Conclusions

- ❖ Three nanostructured equiatomic FeCrCuMnTi, FeCrCuMnTiV, and FeCrCuMnTiVZn HEAs exhibited multiple solid solutions with major FCC phases, minor BCC, and HCP phases evidenced through XRD, HRSEM, and HRTEM.
- ❖ XRD results revealed that FeCrCuMnTiVZn HEA produced more crystallite size reduction, more lattice strain, and a high value of lattice constants due to more structural refinements and high configurational entropy due to incorporation Zn.
- ❖ A greater powder particle size reduction (24.56 ± 1.85 μm) occurred in FeCrCuMnTiV HEA due to the domination of fracturing mechanisms produced by the addition of V atoms.
- ❖ The densification performance indicates that the FeCrCuMnTiV HEA produced a higher relative density of 0.7456 in the as-milled condition and 0.8284 in the stress-recovered conditions. The stress-relieved powder samples exhibited a higher relative density compared to the as-milled powders owing to the elimination of lattice strain.
- ❖ Based on the applied linear and non-linear models, Balshin's Equation (1) and Heckel's Equation (2) linear models are well-fitted linear models. Whereas, Shapiro's Equation (6), and Cooper and Eaton's Equation (7) non-linear models have produced regression coefficients of more than 0.99 indicating the good accuracy of non-linear models compared to linear models.

**Author Contributions:** S.S., conceptualization, methodology, formal analysis, data curation, writing—original draft preparation; H.R.A., conceptualization, formal analysis, writing—review and editing; F.A.A.-M., data curation, resources, writing—review and editing. All authors have read and agreed to the published version of the manuscript.

**Funding:** This research was funded by Qassim University, represented by the Deanship of Scientific Research, grant number (5573-qec-2019-2-2-I).

**Institutional Review Board Statement:** Not applicable.

**Informed Consent Statement:** Not applicable.

**Data Availability Statement:** The experimental datasets obtained from this research work and the analyzed results during the current study are available from the corresponding author on reasonable request.

**Acknowledgments:** The authors gratefully acknowledge Qassim University, represented by the Deanship of Scientific Research, for the financial support for this research under the number (5573-qec-2019-2-2-I) during the academic year 1440 AH/2019 AD.

**Conflicts of Interest:** The authors declare no conflict of interest.

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
