# Peer review of "Influence of V and Zn in FeCrCuMnTi High-Entropy Alloys on Microstructures and Uniaxial Compaction Behavior Prepared by Mechanical Alloying"

_crystals, doi:10.3390/cryst11111413_

Round 1

Reviewer 1 Report

The authors have studied the influence of V and Zn in FeCrCuMnTi high-entropy alloys on microstructures and uniaxial compaction behavior prepared by mechanical alloying. The authors present an extensive study of mechanically alloyed high entropy alloys in this manuscript. The manuscript is well-presented and is suitable for publication.

This reviewer has some suggestions that could be useful for improving the manuscript quality.

The spelling and grammar errors should be checked thoroughly.

Figure 2 (a and b), please check the unit of y-axis, or better keep it “arbitrary units”.

Cantour et al --> Cantor et al.

Schuh et al --> Schuh et al.

The rationale behind the selection of Zn and V should be highlighted in detail. 

The conclusion section is more detailed. Only important points should be highlighted in bullets.

Reviewer 2 Report

The authors have investigated the influence of V and Zn in FeCrCuMnTi high-entropy alloys on microstructures and uniaxial compaction behavior prepared by mechanical alloying. Some good results have been found here. However, there are various issues.

  1. There is a plenty of literature on high entropy alloys. The introduction part needs more recent literature.
  2. Most of the microstructural images are hazy. Please improve the quality of all figures.
  3. EDS compositional analysis should be added for Fig. 3 and 4. The phases should be distinguished.
  4. EDS mapping would be more useful in Fig. 4.
  5. There are many typos “(Error! Reference source not found” which makes difficult to go through entire manuscript.
  6. Page 9: Average particle size is 39.45±2.1 m? Please check the units.
  7. Authors have just presented the observed data but they failed to interpret the results the results in depth. Please improve the discussion as well.

Reviewer 3 Report

This article details an investigation on the densification behavior of three equiatomic high-entropy alloys. It was reported that the stress-recovered powder samples produced more relative density due to the elimination of the lattice strain. A very well put together article that is suitable for the journal, although I have one comment.

1. There were some issues with the references in the manuscript where citation numbers were replaced with the text, "Error! Reference source not found".

Round 2

Reviewer 2 Report

The authors have done a great job in revising the manuscript.